# Comparison of Ultrasonic Phased Array and Film Radiography in Detection of Artificially Embedded Defects in Welded Plates

**DOI:** 10.3390/ma16093579

**Published:** 2023-05-07

**Authors:** Arijan Herceg, Leon Maglić, Branko Grizelj, Vlatko Marušić

**Affiliations:** Mechanical Engineering Faculty, University in Slavonski Brod, 35000 Slavonski Brod, Croatia; arijra@gmail.com (A.H.);

**Keywords:** welded joints, welded plates, volumetric testing, lack of fusion, pores, radiographic testing (RT), ultrasonic testing (UT)

## Abstract

Ultrasonic and radiographic testing are generally two basic methods for volumetric (internal) defect detection in non-destructive testing. Since both methods are commonly used for the same thing, the question arises as to whether both are equally capable of detecting some commonly occurring defects in manufacturing. Commonly occurring defects are generally considered to be fusion defects, drilled holes (which act as pores), etc. To prove or disprove the hypothesis that both methods can generally be used to detect these defects, an experiment was conducted using three welded plates with artificially inserted defects. The welded plates had multiple defects that were intentionally placed close to each other to further complicate the interpretation of the UT results. UT investigation was based on phased-array technology with a multi-element probe. RT investigation was performed with an X-ray machine. Both investigations were based on the respective European standards: for UT, EN ISO 17640, and for RT, EN ISO 17636-1. The results and conclusions from the experiment are presented in this paper.

## 1. Introduction

Detecting discontinuities in welded joints without destroying the joints themselves is an important part of quality control of various fabricated components. Quality control is a series of steps and techniques used to produce end products (welded joints) of satisfactory quality. When considering the quality of welded joints, life cycle and sustainability are of paramount importance [1]. If it is confirmed that the joints are correctly welded, no additional repair work is required, which significantly reduces energy consumption and environmental impact. This is true even when modern, highly efficient welding technologies are used, such as friction stir welding, which by default consumes less energy than conventional gas-shielded metal arc welding. [2]. To confirm that the welds are of acceptable quality, it is necessary to perform examination for detection of surface indications (magnetic testing, penetrant testing, visual testing, eddy current testing) and examination for detection of internal indications. Two basic methods for detecting internal indications are ultrasonic testing (UT) and radiographic testing (RT) [3]. Both UT and RT testing have their own place and purpose based on several factors: material type, material thickness, weld geometry, impact on human health and the environment, costs incurred, etc. [4]. Although they are used for the same purpose, the UT and RT examinations differ in detecting different types of indications. For example, the UT examination is typically used on thicker components to detect indications that are oriented perpendicular to the ultrasound beam (various types of fusion defects). RT can be used on both thicker and thinner components, and it produces the best results when the ionising radiation strikes the target at a 90° angle. Because of the way ionising radiation propagates through matter, there is little theoretical difference when trying to detect spherical indications (pores) or linear indications (melting defects, etc.). UT testing has some clear advantages over RT, such as significant cost reduction and risk reduction of harm to human health and the environment with the use of additional tools such as artificial intelligence [5] or better phased-array imaging [6]. Even with all the guidance provided by referent standards and norms, pertaining to welding and weld examination, it is becoming increasingly difficult to decide between RT and UT. The ultimate goal must be the choice of the optimal testing method for the specific manufactured component (in this case, welded joints) and the creation of a modern system of non-destructive methods [7]. Such a system will help achieve increased quality, safety, and lifespan of manufactured components [8]. The research provided here aims to clarify how efficient each method is in detecting various discontinuities commonly found in welded joints and to help NDT/NDE technologists in the selection of the optimal testing method. Engineering applications of such research can be very important, especially when costs incurred by faulty welding are taken into account. The efficiency of each inspection method is determined by using RT and UT on three welded plates with artificially embedded defects. The embedded flaws are of two types: two-dimensional (planar) fusion flaws and spherical drilled holes. These two types of faults were specifically selected to simulate not only the detectability but also the resolution achieved by each method. Resolution refers to the ability of the methods RT and UT to distinguish between defects that are close in two dimensions. When considering the optimal test parameters for UT and RT testing [9], the authors were guided by the valid standards of EN. For RT testing, EN ISO 17636-1 [10] was relevant, and for UT testing, EN ISO 17640 [11] was relevant. The focus of this work is to study defect detection in a steel alloy for pressure vessels (16Mo3). Further experimentation and consideration is needed concerning the effectiveness of RT and UT for other types of materials.

## 2. Materials and Methods

The experimental tests were performed on welded plates with artificially introduced discontinuities. All three specimens (Specimen No. 1, Specimen No. 2, and Specimen No. 3) were fabricated from 16Mo3, a pressure vessel grade chromium-molybdenum steel alloy defined at EN 10028-2 [12] for use in elevated-temperature environments. Joints made by welding such an alloy should meet all quality requirements and life expectancies specified in the relevant standards, regardless of the welding technique used [13]. Due to its chromium and molybdenum content, 16Mo3 exhibits increased heat and corrosion resistance. The specimens were embedded with discontinuities, some of which were very close in dimensions. An overview of the specimens with their dimensions and the number of embedded discontinuities can be found in Table 1.

Figure 1, Figure 2, Figure 3, Figure 4, Figure 5 and Figure 6 show images of all three specimens (welded plates 1, 2, and 3) and their respective approximate defect locations within the weld material. The specimens were designed to simulate different possibilities for the location of defects relative to each other. For welded plate 1, shown in Figure 1 and Figure 2, the embedded defects were intentionally placed very close to each other. In this way, it was possible to experimentally investigate how such fault positioning affects the test results of UT and RT.

Welded plate 2 and its approximate defect locations are shown in Figure 3 and Figure 4. The positioning of the defects is deliberately different from that of welded plate 1. The defects are relatively far apart, although some are still very close. Such defect placement in the weld should allow realistic testing of the ability of the UT and RT methods to detect and distinguish between different defects. The units in Figure 2, Figure 4 and Figure 6 are in millimetres.

**Figure 3 materials-16-03579-f003:**
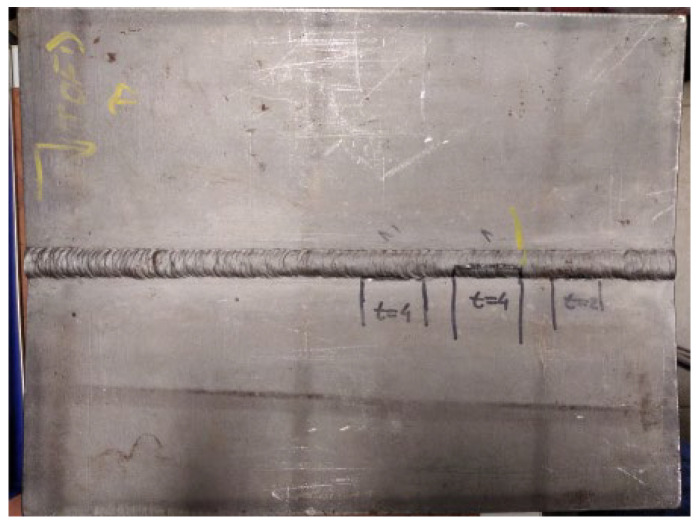
Welded plate—2.

**Figure 4 materials-16-03579-f004:**
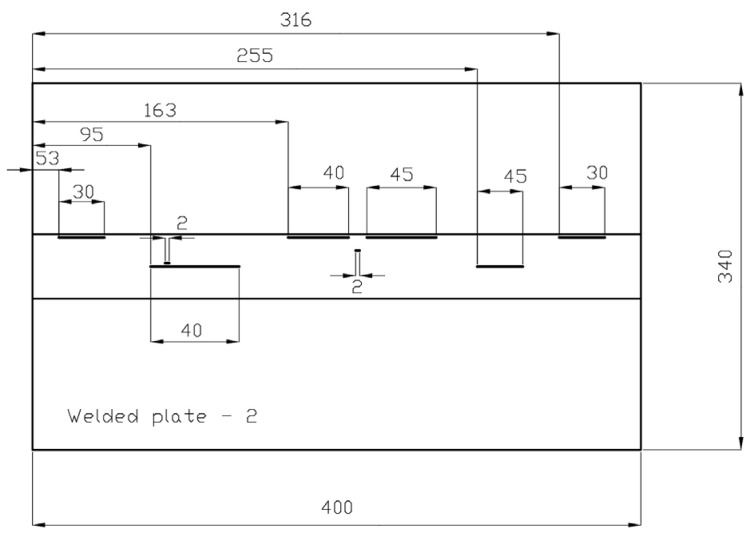
Welded plate—2—approximate defect positions.

The defects in welded plate 3, shown in Figure 5 and Figure 6, were intentionally spaced far apart. With such an arrangement, there should theoretically be no problems with defect detection and the detection of individual defects with either method.

**Figure 5 materials-16-03579-f005:**
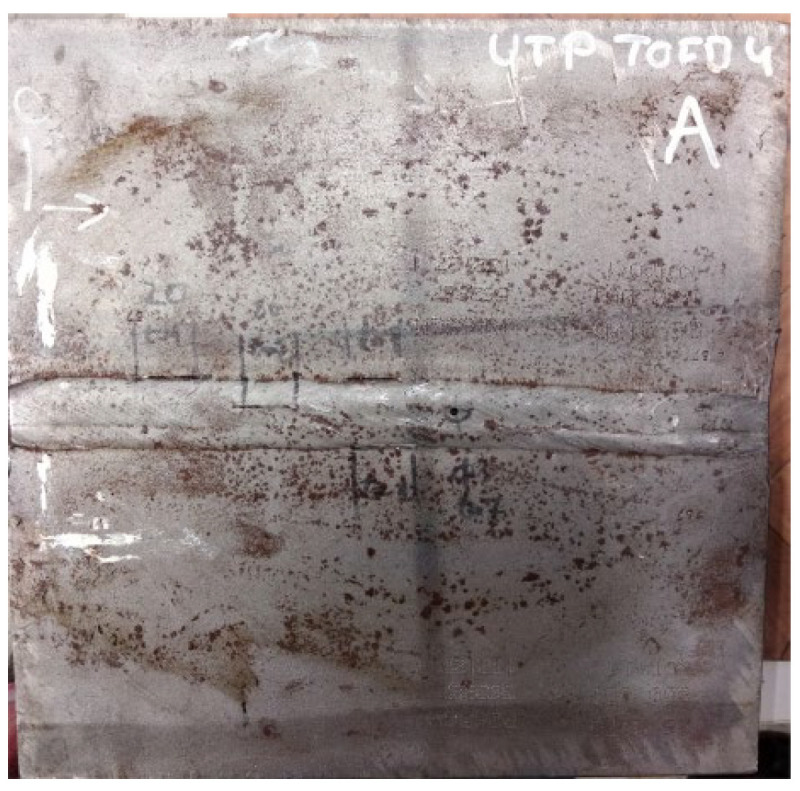
Welded plate—3.

**Figure 6 materials-16-03579-f006:**
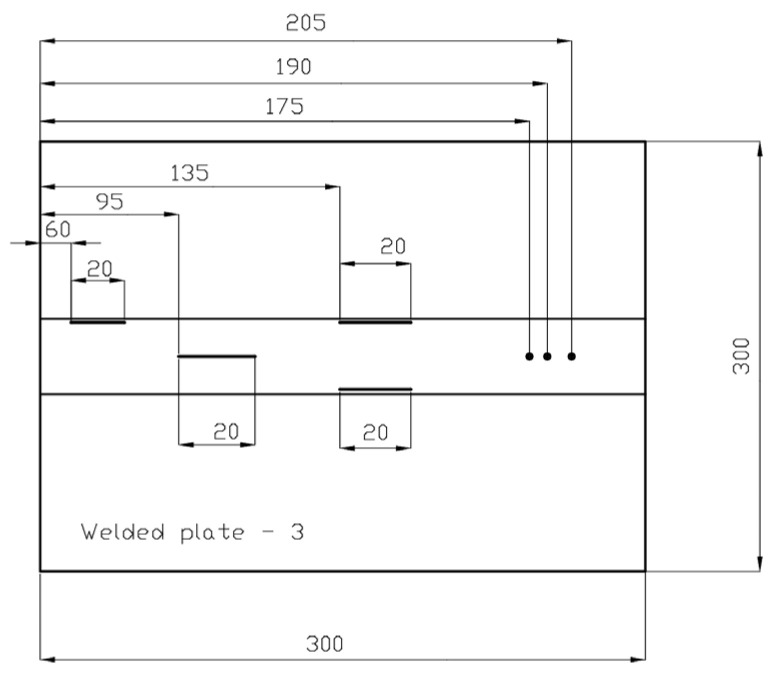
Welded plate—3—approximate defect positions.

The test volume of all three samples was analyzed using the methods RT and UT. The test volume was defined according to EN ISO 17636-1 [10] and EN ISO 17640 [11] as the zone encompassing the weld and base metal and the width of the heat-affected zone on both sides of the weld. An overview of the NDT methods performed is shown in Table 2. The selection of the methods RT and UT was based on the standard EN ISO 17635 [14], shown more precisely in Table 3—Generally accepted methods for detecting internal discontinuities in butt and T-joints with full penetration.

UT examination was performed in accordance with EN IS0 17640 [11] using the following techniques and equipment:UT technique: phased array.UT sensitivity calibration: calibration block + TCG.Phased-array scan: S-scan at fixed probe position with respect to the weld (according to EN ISO 13588).UT device used: Omniscan MX, Olympus.UT probe used: Phased Array: 5L32A1C, SA10-N55S.

RT examination was performed according to EN IS0 17636-1 [10] using the following parameters and equipment:Source type: X-ray device, Eresco 65 MF4, 300 kV/3 mA.RT examination technique: Figure 1.RT films used: 100 × 240, 100 × 480, C3 D4.Target film density: minimum 2.30.RT sensitivity calibration: placement of IQI (Image Quality Indicators) perpendicular to the weld (IQIs are visible on every radiogram presented).

The initial assumption was that using the inspection parameters defined in the reference standards, both RT and UT inspection would provide similar detection and resolution results. All defects present in the welds should be equally detectable and individually determinable with both methods.

## 3. Results

### 3.1. Welded Plate 1

Welded plate 1 had four embedded flaws: three fusion flaws and one drilled hole. All flaws were found on both UT and RT examination. Due to the divergence of the ultrasound beam and the proximity of the flaws to each other, two continuous signals were obtained.

The results of the examination by UT can be seen in Figure 7 and Figure 8 in both the A- and S-scans. The A-scan is the amplitude–time-based part (upper left part of Figure 7 and Figure 8), and the S-scan represents the sectorial part (right part). The UT beam directed into both the weld and the heat-affected zone is reflected from the discontinuities, returns to the probe, and produces a clear and recognizable signal on the screen of the UT instrument. Some variation in accurate defect positioning occurs during UT testing, mainly due to weld geometry and imperfect sound reflection. These factors also contribute to the fact that it was impossible to achieve individual defect resolution. Detected defects generated two continuous signals with small amplitude variations.

Continuous signal 1: fusion flaw (40 mm of length) + fusion flaw (26 mm of length)

**Figure 7 materials-16-03579-f007:**
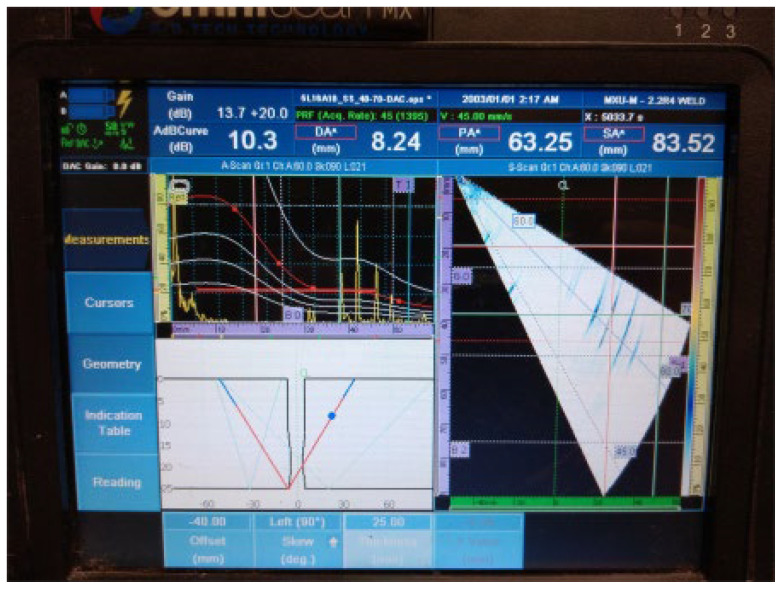
Welded plate—1: continuous signal 1.

2.Continuous signal 2: drilled hole (ϕ2 mm) + fusion flaw (56 mm of length)

**Figure 8 materials-16-03579-f008:**
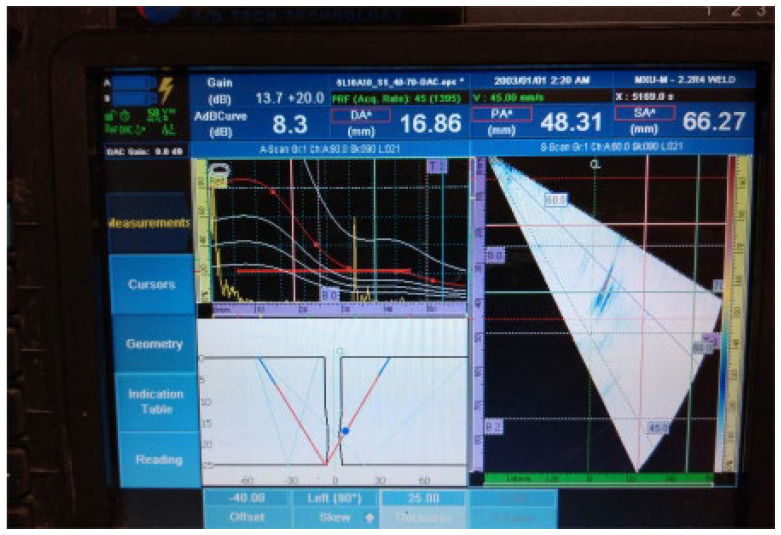
Welded plate—1: continuous signal 2.

Radiograms (Figure 9 and Figure 10) show the results of the examination by RT. The absence of material is shown as a darker area on the film. Due to the nature of the RT examination, the positioning of the defects in two-dimensional space (length and width) is reliable. The X-ray images produced show all four defects individually. Figure 10 shows the characteristic absence of a “white” zone from distribution 20. This is due to the fact that the weld cap was removed by grinding. In this zone, there is less material to absorb the ionizing radiation as it propagates, resulting in a darker radiogram. The difference in optical density can be confirmed both visually and by using a densitometer.

### 3.2. Welded Plate 2

Welded plate 2 had eight embedded flaws: six fusion flaws and two drilled holes. All of the flaws were found during both UT and RT examination. Again, the divergence of the ultrasonic beam and the proximity of the defects to each other resulted in two continuous signals and four individual signals being obtained during the UT examination.

The UT examination results of welded plate 2 (Figure 11, Figure 12, Figure 13, Figure 14 and Figure 15) show the detected defects and their approximate positions within the test zone. Similar to the results of welded plate 1, both the amplitude in the A-scan and the differently coloured area in the sectorial scan indicate the presence of defects. If necessary, by changing the position of the probe UT during the test and by increasing the gain, an even stronger sound reflection and thus easier detection can be achieved. However, even with these parameters selected, results are easily obtained.

Individual signal 1: fusion flaw (30 mm of length)

**Figure 11 materials-16-03579-f011:**
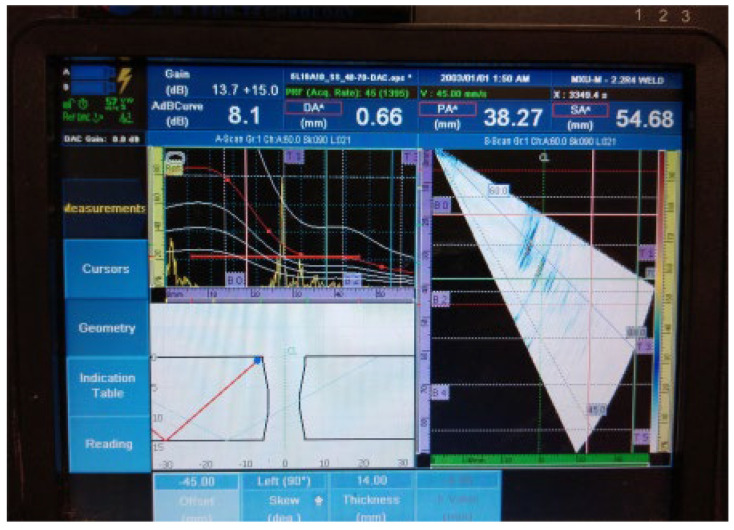
Welded plate—2: individual signal 1.

Spherical defects (drill holes), as seen in Figure 12, generally produce lower acoustic energy reflection and are therefore more difficult to detect than areal defects (lack of fusion). This can be easily verified by following the amplitude peak in the A-scan (upper left part of the UT instrument screen). The amplitude drop in the detection of spherical defects was 40% of the screen height, as confirmed by a comparison of Figure 11 and Figure 12.

2.Continuous signal 1: drilled hole (ϕ2 mm) + fusion flaw (40 mm of length)

**Figure 12 materials-16-03579-f012:**
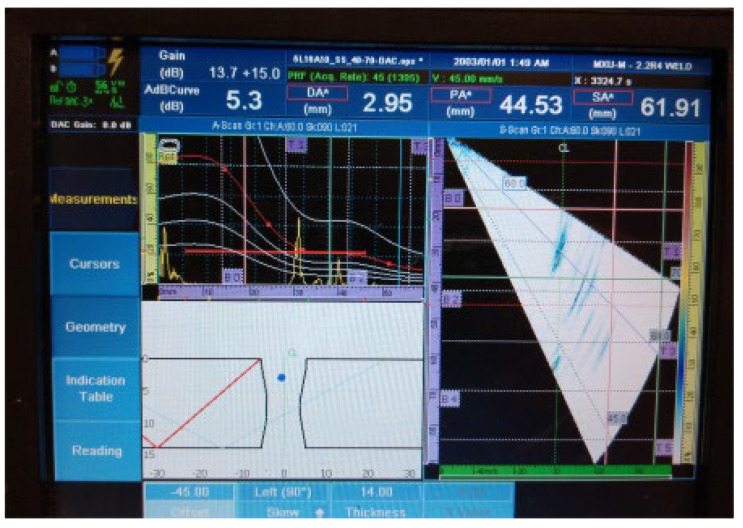
Welded plate—2: continuous signal 1.

The accuracy of defect positioning, visible in the lower left part of the screen of the UT device on all images for UT testing, depends on the quality of data available to the inspector and configured in the UT device. Weld type, dimensions, bevel geometry, and material must be correctly configured for defect positioning to allow accurate defect location. This is also the main reason why some discrepancies exist between the actual defect position and the displayed defect position.

3.Continuous signal 2: drilled hole (ϕ2 mm) + fusion flaw (40 mm of length) + fusion flaw (45 mm of length)

**Figure 13 materials-16-03579-f013:**
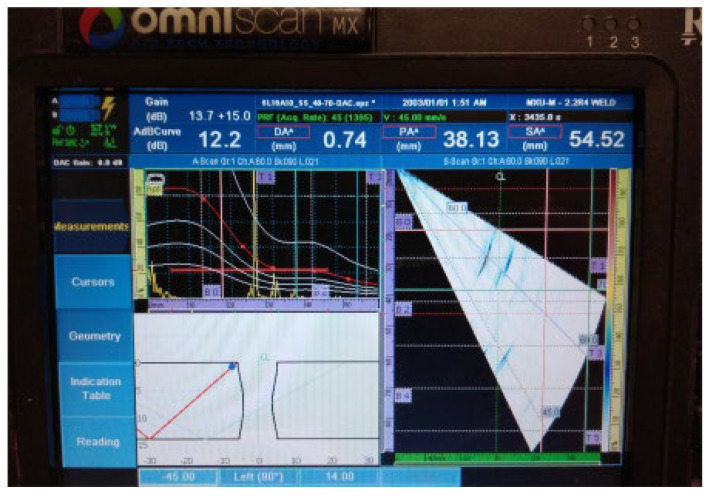
Welded plate—2: continuous signal 2.

4.Individual signal 2: lack of fusion (45 mm of length)

**Figure 14 materials-16-03579-f014:**
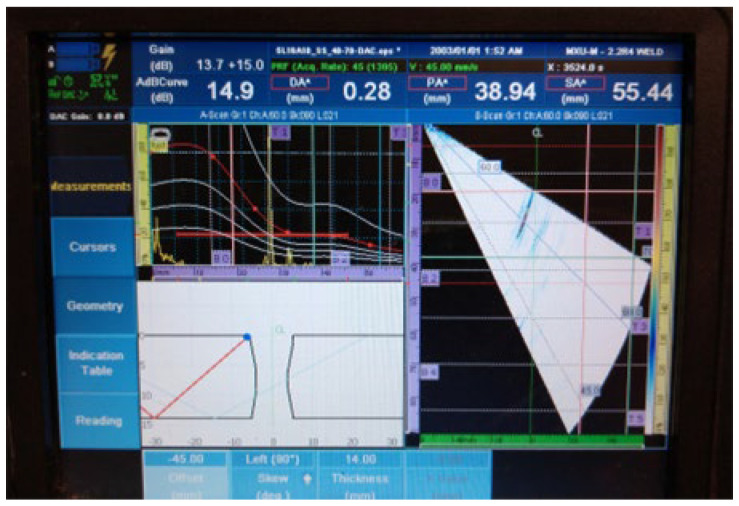
Welded plate—2: individual signal 2.

5.Individual signal 3: lack of fusion (30 mm of length)

**Figure 15 materials-16-03579-f015:**
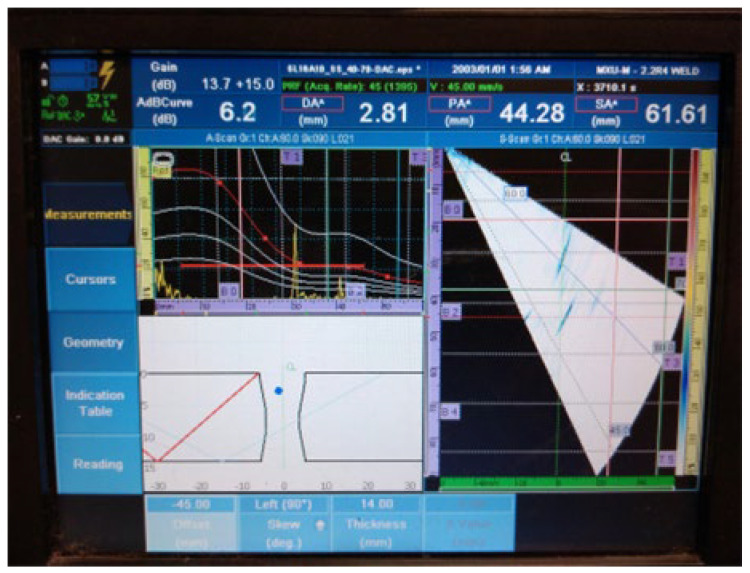
Welded plate—2: individual signal 3.

Following the results of welded plate 1, it was not possible to achieve complete resolution of the individual defects. Continuous signals clearly indicate the presence of defects, but UT cannot distinguish between two (or more) closely spaced defects.

Figure 16 and Figure 17 show the results of RT testing for welded plate 2. Two radiograms were required to cover the entire area examined. Defects are visible as darker areas in the white of the weld and base metal. Both radiograms also show how some relatively close defects can produce a continuous signal when examined at UT and appear as a single defect (continuous signal 2).

### 3.3. Welded Plate 3

Welded plate 3 had seven embedded flaws: four fusion flaws and three drill holes. All flaws were found during both UT and RT inspection. In the UT inspection of welded plate 3, it was possible to achieve individual resolution of the defect signals.

Figure 18, Figure 19, Figure 20, Figure 21, Figure 22, Figure 23 and Figure 24 show the results of the test UT. For welded plate 3, all embedded defects were relatively far apart or were located in different weld sections, so it was possible to identify individual defect signals. The defects were oriented and large enough to allow significant ultrasonic reflection. The sound returning to the phased-array probe resulted in a significant amplitude that was visible on the amplitude–time-based portion of the UT instrument screen. The sectorial scan also confirmed the survey results, as it was based on the A-scan.

Individual signal 1: fusion flaw (20 mm of length)

**Figure 18 materials-16-03579-f018:**
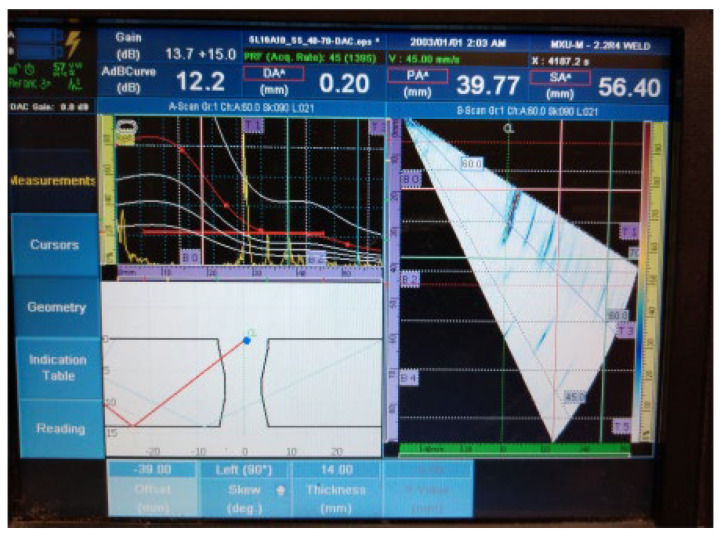
Welded plate—3: individual signal 1.

2.Individual signal 2: fusion flaw (20 mm of length)

**Figure 19 materials-16-03579-f019:**
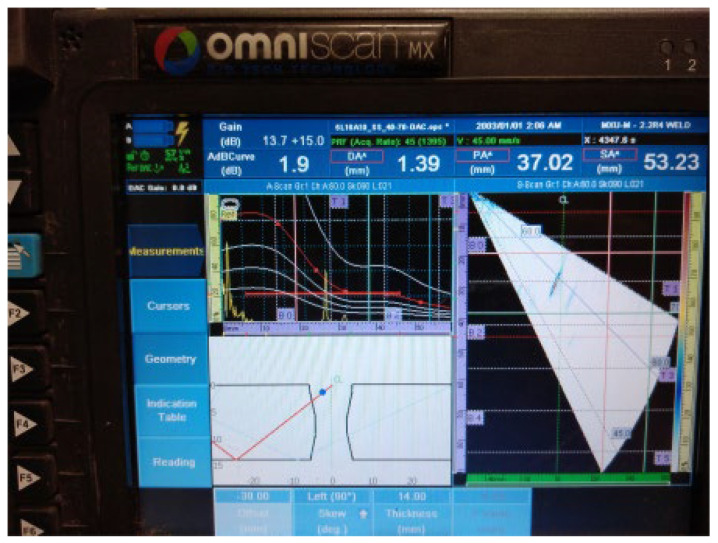
Welded plate—3: individual signal 2.

3.Individual signal 3: fusion flaw (20 mm of length)

**Figure 20 materials-16-03579-f020:**
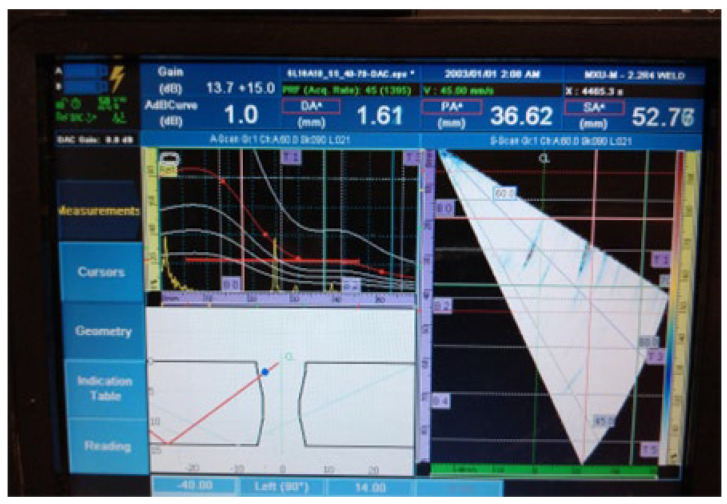
Welded plate—3: individual signal 3.

4.Individual signal 4: fusion flaw (20 mm of length)

**Figure 21 materials-16-03579-f021:**
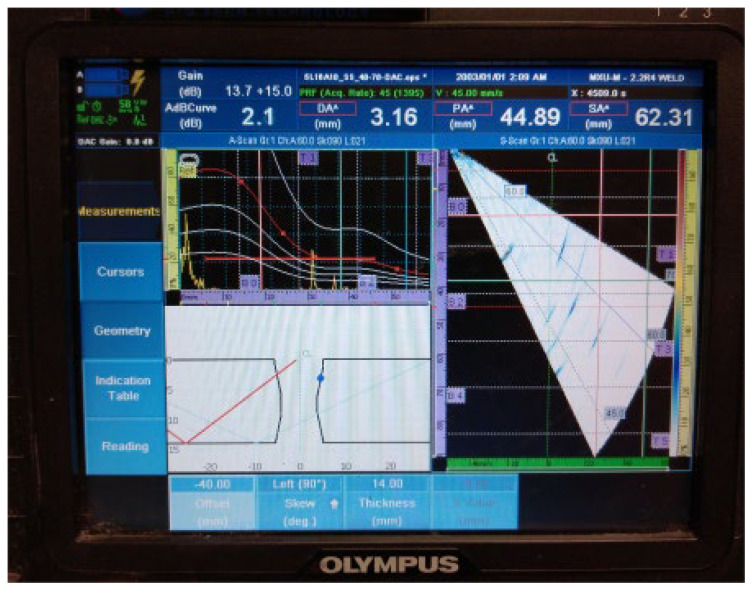
Welded plate—3: individual signal 4.

The approximate location of the flaw in the weld area, shown in the lower left portion of Figure 18, Figure 19, Figure 20, Figure 21, Figure 22, Figure 23 and Figure 24 also achieved acceptable accuracy. This is especially important because there are significant differences between the ideal weld configuration available in the instrument UT and the actual weld configuration in the welded plate. Sound reflection must also be taken into account when determining the UT defect location. Since it was possible to obtain individual signals if needed, the approximate defect size could also be determined by the amplitude decay technique. A fusion flaw in the centre of the weld (individual signal 2) could also be effectively used to simulate a longitudinal crack.

5.Individual signal 5: drilled hole (ϕ2 mm)

**Figure 22 materials-16-03579-f022:**
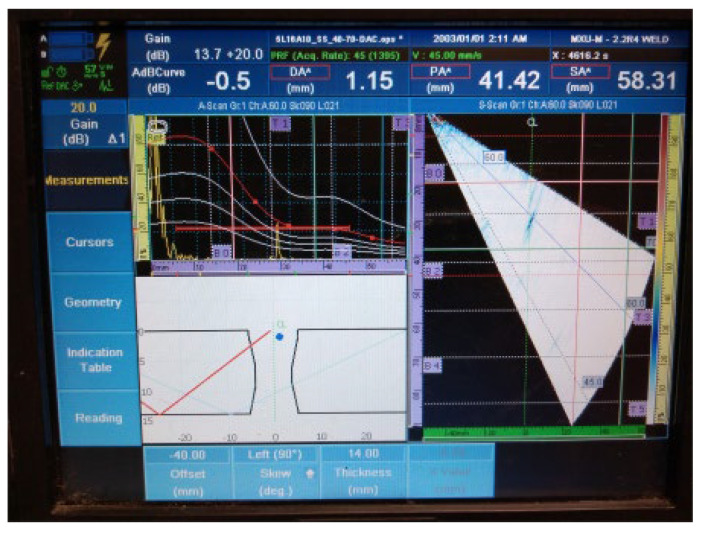
Welded plate—3: individual signal 5.

6.Individual signal 6: drilled hole (ϕ2 mm)

**Figure 23 materials-16-03579-f023:**
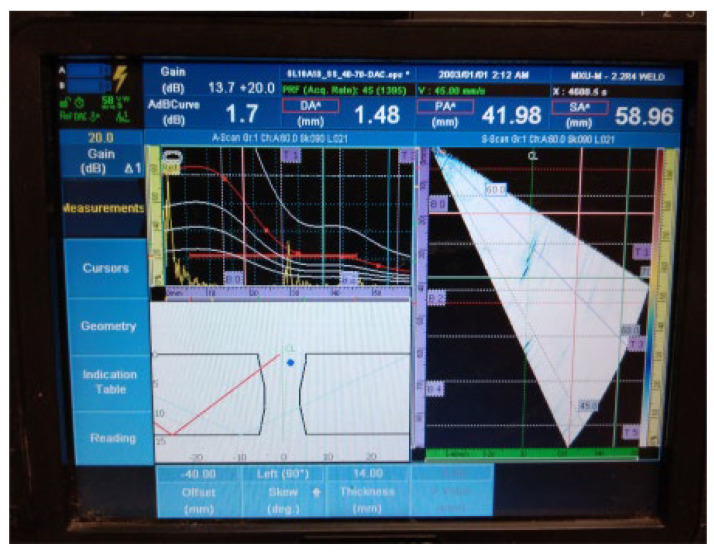
Welded plate—3: individual signal 6.

7.Individual signal 7: drilled hole (ϕ2 mm)

**Figure 24 materials-16-03579-f024:**
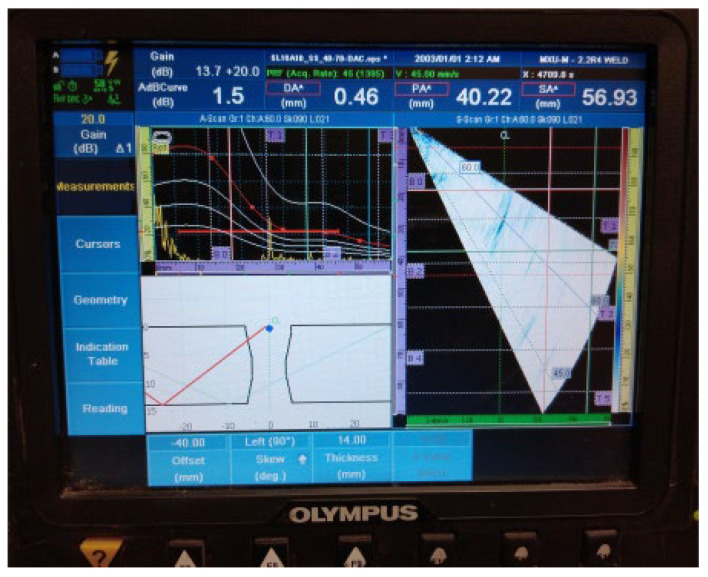
Welded plate—3: individual signal 7.

The radiogram for welded plate 3 (Figure 25) also clearly shows all defects. According to the principle of different optical density, the defects are shown as dark areas in the white of the weld. The spatial distribution of the defects within the weld explains why it was possible to detect individual defect signals with UT. As with earlier plates, classical RT testing can reliably determine defect size in two dimensions (length and width). The third dimension, which can be understood as both defect depth and depth in the weld metal, cannot be reliably determined. This is due to the technical limitations of the RT method itself.

## 4. Discussion

According to Table 3, all defects were successfully detected using either the UT or the RT test. The UT test detected both fusion defects and borehole defects, but it failed to detect any individual defects. The RT test also detected all defects, but at the same time, it achieved clear individual defect detection.

Table 4 shows the results of the UT and RT tests of welded plate 2. From Table 4, it can be seen that all defects could be detected in the UT test. Furthermore, UT was able to detect individual defects for defects 1, 7, and 8. RT was not able to detect the defects or distinguish them individually.

The results of the examination of welded plate 3 are shown in Table 5. According to Table 5, all embedded defects were successfully detected by both methods. The main difference with the investigation results of welded plates 1 and 2 is that individual defect detection is obtained for all embedded defects using both UT and RT. The reasons for these results will be given later in the discussion.

The results discussed in this paper were obtained by double-testing (UT and RT) three welded plates. The plates were embedded with artificial defects, mainly fusion flaws and drilled holes. These are not, of course, the only defects that can occur during welding, but they are among the most common and are therefore relevant to the discussion. Boreholes are used to simulate different types of porosity because of their spherical shape. Depending on their size and orientation with respect to the ultrasonic beam, they can be extremely difficult to detect with UT. A lack of fusion, especially if oriented perpendicular to the UT sonic beam, is much easier to detect. RT examination is also most effective when the ionising radiation is directed perpendicular to the defect, but it is quite effective at detecting all types of discontinuities. Testing with UT and confirming the results with another volumetric method (RT) provide the basis for comparing the detection capabilities of the two methods. The investigation was performed with the phased-array method UT and the classical method RT. The results of the experiments show that both investigation methods successfully detected all embedded defects. Differences between the methods RT and UT become apparent when the defect resolution is taken into account. The ability of the UT method to correctly detect individual defects is very different from the RT method. The main reason for this is the spatial and dimensional position of the defects in relation to each other, as shown in Figure 26. In other words: When two defects are close to each other, it is very difficult to determine their correct number. The threshold value for the distance between two defects that can still be recognised as individual has been experimentally set at 15 mm. This value is based on experiments with the specific UT setup, mainly UT device and phased-array probe. Better results may be obtained with other UT setups.

The aim of this study is to provide an experimental reference for situations where it is necessary to decide on the use of a particular technique when detecting volumetric defects.

## 5. Conclusions

The examination results of welded plates 1, 2, and 3 confirm the assumption that it is possible to identify all embedded defects with both UT and RT. Table 3, Table 4 and Table 5 show that the main difference between the results obtained with RT and UT is the detection of individual defects. Experimental investigation confirmed that an approximate minimal distance between indications of 15 mm is necessary for UT to produce individual signal detection.

The experimental investigation resulted in the following conclusions:UT examination has detected all embedded defects.RT examination has detected all embedded defects.In some cases, the UT examination is not able to distinguish between two closely spaced defects.RT examination has detected all embedded defects individually.The use of RT or UT depends on several factors: dimensions and geometry of the part under study, manufacturing process of the part under study, typical defects expected for the selected manufacturing process, constraints imposed by human health and environmental concerns, etc.

## Figures and Tables

**Figure 1 materials-16-03579-f001:**
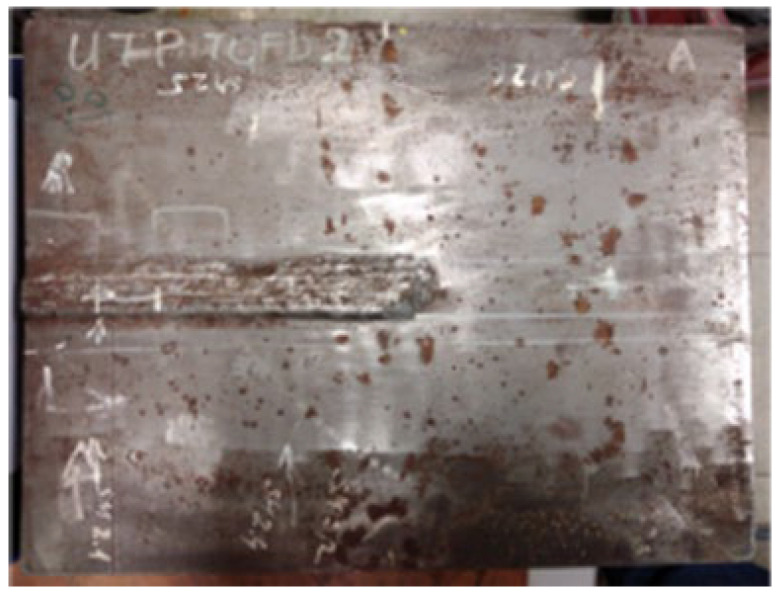
Welded plate—1.

**Figure 2 materials-16-03579-f002:**
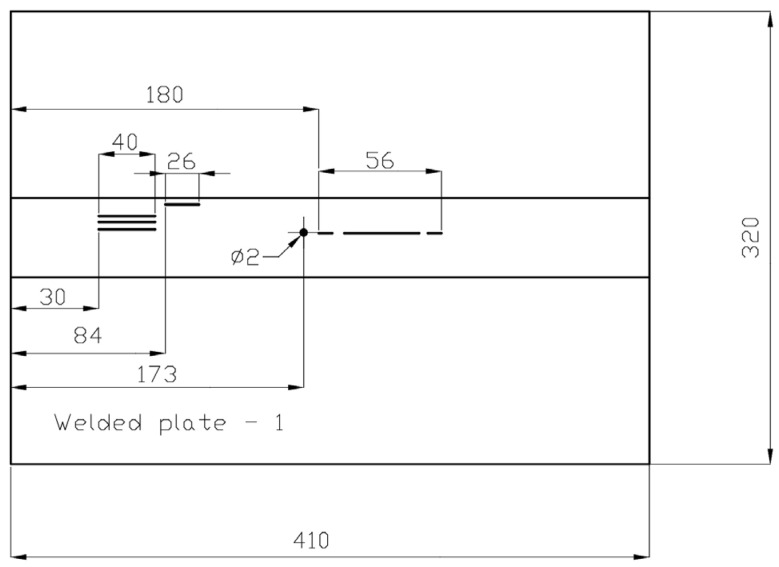
Welded plate—1—approximate defect positions.

**Figure 9 materials-16-03579-f009:**
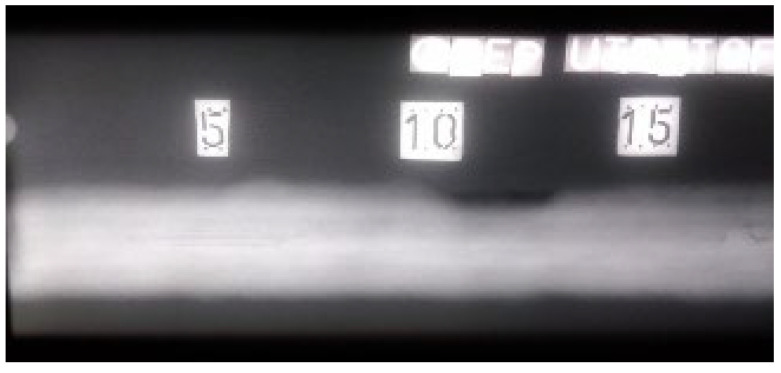
Radiogram of welded plate 1: 0–15 distribution.

**Figure 10 materials-16-03579-f010:**
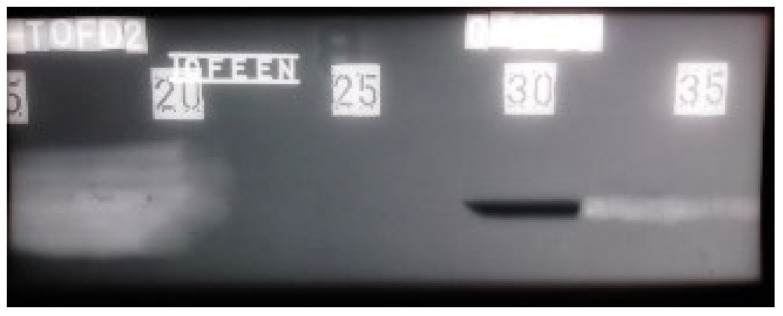
Radiogram of welded plate 1: 15–35 distribution.

**Figure 16 materials-16-03579-f016:**
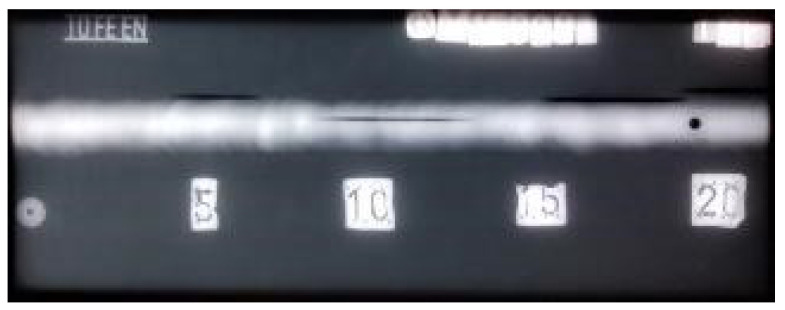
Radiogram of welded plate 2: 0–20 distribution.

**Figure 17 materials-16-03579-f017:**
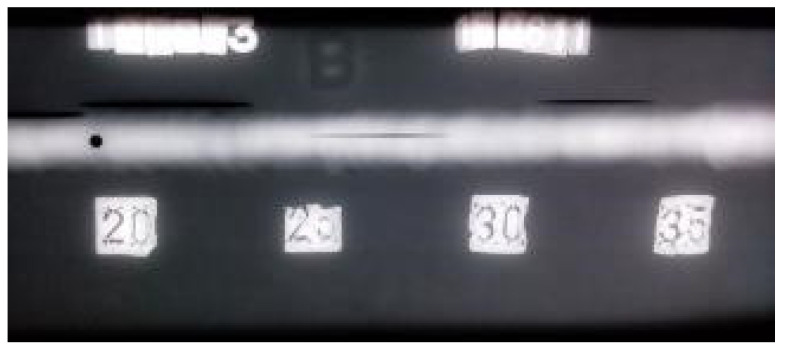
Radiogram of welded plate 2: 20–35 distribution.

**Figure 25 materials-16-03579-f025:**
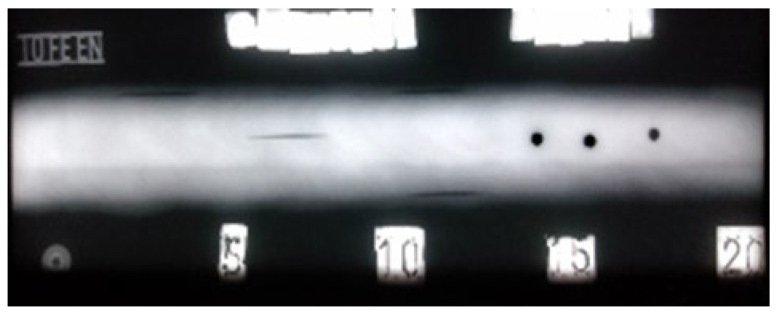
Radiogram of welded plate 3: 0–20 distribution.

**Figure 26 materials-16-03579-f026:**
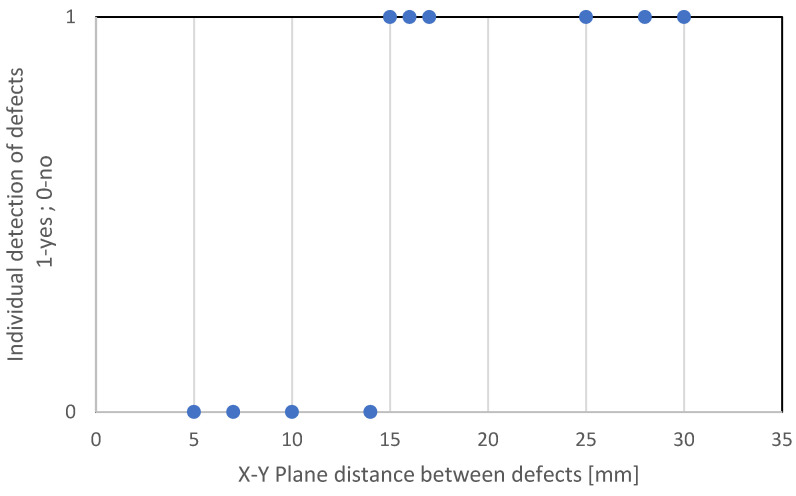
Dependence of individual detection of defects during UT examination on the distance between defects.

**Table 1 materials-16-03579-t001:** Examined plates/samples.

Sample No.	Part Description	Material	Dimensions mm	No. of Discontinuities
1	Welded plate	16Mo3	320 × 410 × 25	4
2	Welded plate	16Mo3	340 × 400 × 14	8
3	Welded plate	16Mo3	300 × 300 × 15	7

**Table 2 materials-16-03579-t002:** Performed NDT.

Sample No.	Weld Type	RT—Testing Volume	UT—Testing Volume
1	butt: plate/plate	weld + parent material + heat-affected zone	weld + parent material + heat-affected zone
2	butt: plate/plate	weld + parent material + heat-affected zone	weld + parent material + heat-affected zone
3	butt: plate/plate	weld + parent material + heat-affected zone	weld + parent material + heat-affected zone

**Table 3 materials-16-03579-t003:** Examination results of welded plate 1.

Welded Plate 1					
No. of Defect	Defect Type	Detected by UT	Detected by UT as Individual Defect	Detected by RT	Detected by RT as Individual Defect
1	Fusion flaw	yes	no	yes	yes
2	Fusion flaw	yes	no	yes	yes
3	Drilled hole	yes	no	yes	yes
4	Fusion flaw	yes	no	yes	yes

**Table 4 materials-16-03579-t004:** Examination results of welded plate 2.

Welded Plate 2					
No. of Defect	Defect Type	Detected by UT	Detected by UT as Individual Defect	Detected by RT	Detected by RT as Individual Defect
1	Fusion flaw	yes	yes	yes	yes
2	Drilled hole	yes	no	yes	yes
3	Fusion flaw	yes	no	yes	yes
4	Fusion flaw	yes	no	yes	yes
5	Drilled hole	yes	no	yes	yes
6	Fusion flaw	yes	no	yes	yes
7	Fusion flaw	yes	yes	yes	yes
8	Fusion flaw	yes	yes	yes	yes

**Table 5 materials-16-03579-t005:** Examination results of welded plate 3.

Welded Plate 3					
No. of Defect	Defect Type	Detected by UT	Detected by UT as Individual Defect	Detected by RT	Detected by RT as Individual Defect
1	Fusion flaw	yes	yes	yes	yes
2	Fusion flaw	yes	yes	yes	yes
3	Fusion flaw	yes	yes	yes	yes
4	Fusion flaw	yes	yes	yes	yes
5	Drilled hole	yes	yes	yes	yes
6	Drilled hole	yes	yes	yes	yes
7	Drilled hole	yes	yes	yes	yes

## Data Availability

The data presented in this study are available on request from the corresponding author. The data are not publicly available due to company regulations.

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
