# Peer review of "Comparison of Ultrasonic Phased Array and Film Radiography in Detection of Artificially Embedded Defects in Welded Plates"

_materials, 2023, doi:10.3390/ma16093579_

Round 1

Reviewer 1 Report

Ultrasonic and radiographic testing are generally two basic methods for volumetric (in- 7ternal) defect detection in nondestructive testing. Since both methods are colloquially used for the 8same thing, the question arises as to whether both are equally capable of detecting some commonly occurring defects in manufacturing. Commonly occurring defects are generally considered to be fusion defects, drilled holes (which act as pores), etc. To prove or disprove the hypothesis that both methods can generally be used to detect these defects, an experiment was conducted using three 12welded plates with artificially inserted defects. The welded plates had multiple defects that were 13 intentionally placed close to each other to further complicate the interpretation of the UT results. 14 UT Investigation was based on phased array technology with multi-element probe. RT Investigation was performed with an X-ray machine. Both investigations were based on the respective European 16standards for UT EN ISO 17640 and for RT EN ISO 17636-1. The results and conclusions from the 17 experiment were presented in Chapters 3, 4 and 5. 

1.The English usage and grammar in this manuscript were checked, there are minor grammatical errors throughout the whole manuscript.

2. In the introduction, some literatures are simply listed. A brief review with logical depth is required to emphasis the research significance of this work. The research significance of this manuscript is need re-considered not only by the view of fundamental research but also by the perspective of engineering application.

3.Into 2. Materials and methods

Kindly change “METHODOLOGY” to “2. Materials and methods

4.Into Results

In line 143, Kindly change “UT the examination results” to “the examination results UT”

5.Into Discussion

Please expand this section according to Table 3 to Table 5.

6.Into Conclusions

The conclusion part should be concise and concrete.

Three or four paragraphs describing the main findings of this study should be given before the bullet points of conclusions.

7.Please updates the references for the year 2022 and the year 2023.

Author Response

  1. English text checked by software Instatext

  1. The introduction has been expanded to explain the importance of the research and the potential benefits for modern welding techniques.

  1. “METHODOLOGY” changed to “ Materials and methods”

  1. In line 143, Kindly change “UT the examination results” to “the examination results UT”

Text have been changed line 163

5.Into Discussionall

Please expand this section according to Table 3 to Table 5.

Discussion text  is added to explain data in tables

6.Into Conclusions

Conclusion text is changed

7.Please updates the references for the year 2022 and the year 2023.

References were updated

Reviewer 2 Report

The article compares the effectiveness of ultrasonic testing (UT) and radiographic testing (RT) for detecting internal defects in welded joints.

The article provides a clear introduction to the importance of non-destructive testing and the differences between UT and RT. The methodology section is detailed, including the selection of materials and test equipment, and the results are presented in a well-organized manner. However, the differences between the three different investigated plates are not explained. Where were the defects positioned?

The literature survey on the two types of tests is not completely exhaustive. This is not acceptable for a document whose main purpose is to present a comparison of methodologies. The authors are invited to include some relevant references on the topic. In addition, given the ever-increasing attention towards the theme of sustainability, it is recommended to include some considerations. i.e the following references could be used to expand the bibliography also on the issue concerning life cycle assessment [R1-R4] and give some considerations.

[R1]       Integrating the sustainability aspects into the risk analysis for the manufacturing of dissimilar aluminium/steel friction stir welded single lap joints used in marine applications through a Life Cycle Assessment,  Sustainable Futures, Volume 4, 2022

[R2]       Life Cycle Assessment of welding technologies for thick metal plate welds, Journal of Cleaner Production, Volume 108, Part A, 1 December 2015, Pages 46-53

[R3]       Life cycle analysis for laser welding of alloys,Optics & Laser Technology, Volume 126, June 2020

[R4]       Comparison of energy consumption and environmental impact of friction stir welding and gas metal arc welding for aluminum, CIRP Journal of Manufacturing Science and Technology, Volume 9, May 2015, Pages 159-168

Not all the figures are properly introduced within the text. Furthermore, the text should be placed before figures.

Figures concerning the signal acquired (i.e., Fig 4) are all bad quality and not properly explained. Please provide.

Overall, the study provides some good information for quality control of welded components. The article is well written, easy to understand, and presents relevant information.

The limitations of the study are not discussed, and future research may explore the effectiveness of UT and RT methods for other types of defects and materials. Some consideration should be given.

Author Response

  1. However, the differences between the three different investigated plates are not explained. Where were the defects positioned?

 Figures 2,4 and 6 were added to define defect positions

  1. References were updated

  1. Not all the figures are properly introduced within the text. Furthermore, the text should be placed before figures.

Text for figures is added (i.e Fig. 12 and Fig 13)

  1. Figures concerning the signal acquired (i.e., Fig 4) are all bad quality and not properly explained. Please provide.

Text added at lines 172-179  and 183-189

Reviewer 3 Report

Very good professional article, but with a low scientific level.

The resolution was not investigated, resolution test pattern were not used in case of RT, sensitivity and resolution are two terms that must be investigated in UT.

The article cannot be considered scientific.

Author Response

The resolution was not investigated, resolution test pattern were not used in case of RT, sensitivity and resolution are two terms that must be investigated in UT.

UT sensitivity mentioned at line 105

Round 2

Reviewer 1 Report

1. In the introduction, some literatures are simply listed. A brief review with logical depth is required to emphasis the research significance of this work. The research significance of this manuscript is need re-considered not only by the view of fundamental research but also by the perspective of engineering application.

2.Into Conclusions

The conclusion part should be concise and concrete.

Three or four paragraphs describing the main findings of this study should be given before the bullet points of conclusions.

Into Conclusions

The conclusion part should be concise and concrete.

Three or four paragraphs describing the main findings of this study should be given before the bullet points of conclusions.

Author Response

1. 

Dear reviewer, we have added additional lines of text in rows 23, 25, 30-34, 43-50, which hopefully address the issues you had with Introduction written the way it was before. Literature has been added in this manner to highlight all the reasons why quality control is so important and why determining acceptable and unacceptable welds has such a large impact on life, environment, and cost. The main technical application of this work is as a potential aid to other NDT/NDE technologists in planning and conducting practical testing in their own industries. We also hope that we have emphasised this in our work. We thank you for your feedback.

2. 

We have additionally revised the Conclusions chapter by deleting unnecessary parts and adding some new, more concise ones: 314-317,329-330, 334, 336. We hope this is to your liking.

Reviewer 2 Report

The authors addressed the reviewer's comments properly.

Author Response

Thank You

Reviewer 3 Report

Without determining the resolution of the methods, it is impossible to compare them. Resolution test pattern must be used in case of RT, it was not done.

Author Response

Dear reviewer, if our understanding of the resolution test pattern for RT is correct, we would humbly like to draw your attention to the next sentences. Resolution and sensitivity for testing UT were achieved by making a calibration block of the same material with SDH (side drilled holes) and notches required by the standard. UT was calibrated on this block by performing TCG (Time Corrected Gain). The sensitivity or resolution (i.e., how small the indications are that we can detect) on the X-ray images is confirmed by placing an IQI (image quality indicator) perpendicular to the weld on all X-ray images presented in this paper. These IQI-s consist of wires of different thickness, and which wire must be seen is specified in the standard for testing RT (EN ISO 17636-1). When the radiograph images are enlarged, the IQI-s are clearly visible and the required wires are also clearly visible. Just to emphasise how the system of resolution definition works with the help of IQI-s, the thinnest required wire must be seen on each radiograph in a length of at least 10 mm on the surface with uniform thickness. We are sure that you will also agree that this requirement is clearly met in all radiographs presented here. If we have misunderstood your objection in any way, we sincerely apologise. To clarify this, we have added an additional line of text to lines 123 and 124 in our paper.

Round 3

Reviewer 1 Report

The English usage and grammar in this manuscript were checked, a native English speaker or an experienced proofreader are recommended for proofreading the manuscript. 

Reviewer 3 Report

In my opinion, in the case of RT the resolution  must be examined  using the resolution test pattern.